# Damage dynamics and the role of chance in the timing of *E. coli* cell death

Yifan Yang ®[1,2] ✉, Omer Karin[1], Avi Mayo ®[1], Xiaohu Song[2], Peipei Chen ®[2,3], Ana L. Santos[2,4], Ariel B. Lindner ®[2] & Uri Alon ®[1] ✉

Genetically identical cells in the same stressful condition die at different times. The origin of this stochasticity is unclear; it may arise from different initial conditions that affect the time of demise, or from a stochastic damage accumulation mechanism that erases the initial conditions and instead amplifies noise to generate different lifespans. To address this requires measuring damage dynamics in individual cells over the lifespan, but this has rarely been achieved. Here, we used a microfluidic device to measure membrane damage in 635 carbon-starved *Escherichia coli* cells at high temporal resolution. We find that initial conditions of damage, size or cell-cycle phase do not explain most of the lifespan variation. Instead, the data points to a stochastic mechanism in which noise is amplified by a rising production of damage that saturates its own removal. Surprisingly, the relative variation in damage drops with age: cells become more similar to each other in terms of relative damage, indicating increasing determinism with age. Thus, chance erases initial conditions and then gives way to increasingly deterministic dynamics that dominate the lifespan distribution.

Genetically identical organisms placed in the same conditions die at different times[1–5]. This non-genetic variation is shared also by single-celled organisms, such as starving *Escherichia coli* (*E. coli*)[4] and aging yeast[6].

Two possibilities have been raised to understand this stochasticity of death times[1,2,7,8]. The first is that the initial states of individuals are different and affect the eventual time of demise[2,8]. The second is that initial conditions are rapidly erased by stochastic accumulation of damage over time, and stochasticity further accumulates to cause the different lifespans[7]. The nature of this stochastic accumulation is unclear.

To understand the role of chance and initial conditions in the timing of cell death, it is essential to measure overtime the damage that causes death in individual cells. This, however, has rarely been done.

Here we use carbon-starved *E. coli* in microfluidic chambers to study the role of stochasticity and initial conditions in the time of cell death. The cells have a risk of death that rises exponentially with age[4], known as the Gompertz law, which also characterizes mortality in other microorganisms and animals[9]. We use the well-established bacterial viability marker propidium iodide[10] to measure membrane damage in individual cells in the microfluidic device. We find that initial conditions of damage or cell-cycle phase do not strongly correlate with time of death. Instead, the data suggests a specific mechanistic model for the stochastic dynamics of the damage that causes death. In this model, damage-producing units such as unfolded protein complexes rise at a constant rate and produce damage, whose removal processes saturate at high damage levels. This saturation amplifies noise and leads to different individual dynamics, explaining the majority of variation in lifespan. Surprisingly, the relative damage variation among cells drops with age, indicating that stochasticity erases initial conditions, but then becomes less dominant and damage dynamics becomes increasingly deterministic with age.

[1]Department of molecular Cell biology, Weizmann Institute of Science, 71600 Rehovot, Israel. [2]Université de Paris - INSERM Unit 1284, Center for Research and Interdisciplinarity (CRI), Paris F-75004, France. [3]Present address: National Center for Nanoscience and Technology, 100190 Beijing, China. [4]Present address: Department of Chemistry, Rice University, Houston, TX 77005, USA. ✉e-mail: yifan.yang@weizmann.ac.il; uri.alon@weizmann.ac.il

## Results

### *E. coli* damage dynamics in individual cells

We tracked individually starved *E. coli* cells by time-lapse microscopy using a microfluidic system called the mother machine[11] (Fig. 1a, Supplementary Fig. 1, Supplementary Movie 1). Individual cells from the same clone were loaded onto an array of dead-end micro-channels (6 μm long and 1.1 μm high and wide) that open onto a main channel[12]. The micro-channels exposed the individual cells to a homogenous medium refreshed by flow in the main channel in which the cells were starved for carbon.

The device prevented cells from interacting. This bypasses the effects of feeding on the remains of perished cells that occur in batch culture starvation which lead to an exponential survival curve with a constant risk of death[13] rather than the Gompertz law observed in the microfluidic device[4].

To allow different initial cell-cycle phases and cell sizes, we loaded the cells onto the chip from a culture in exponential growth. Thus, some cells have recently divided whereas others are about to divide. The chip was then thoroughly washed to eliminate traces of carbon nutrient[4].

To follow the physiological deterioration process of each cell, we focused on membrane integrity as an indicator of damage. Membrane integrity is critical to a cell's survival[14–16] and is affected by many physiological parameters, including pH, redox balance, energy metabolism and translation fidelity[17].

We measured membrane integrity with propidium iodide (PI), a well-established non-toxic dye for bacterial viability[4]. PI becomes fluorescent only when it penetrates the cell membrane and binds to DNA. Due to its relatively large size and charge, PI cannot cross the membrane when the membrane is functionally intact. We therefore used the rate of PI uptake to quantify the integrity of bacterial membranes (Fig. 1b). PI uptake rate was calculated from the image time-series of each bacterium at resolution of 1 h (Methods). Experimental noise of the fluorescence image time-series is estimated at about 6% (Supplementary Fig. 2). PI-base viability in the assay compared well with another viability stain which has a distinct, protein-based mechanism (Supplementary Fig. 3).

According to the Arrhenius equation, PI uptake rate is inversely proportional to the exponential of the potential barrier that the PI molecule has to cross to enter the cell. We therefore define membrane damage $X(t)$ as the log of the PI uptake rate normalized to the mean uptake rate of the initial population (Methods). Cell death was determined by damage levels exceeding a threshold, $X_c$. The value of $X_c$ is determined by the maximal $X(t)$ observed before cells reach previously established lifespans[4].

Cells survived for an average of 82 h (Fig. 1c), and showed an exponentially rising risk of death (Fig. 1e), namely the Gompertz law[4]. Cells rarely die in the first 40 h, and then begin to die more and more frequently, leading to a sigmoidal survival curve. The relative variation of death times was 24%, where 5% of the cells died by 42 h, and 95% died by 106 h.

From the time-series of PI fluorescence we measured the damage $X(t)$ in 635 individual bacterial cells at 8 time points, which correspond to 8 non-overlapping windows of 7 h each between 20 h and 80 h (Fig. 1b–d). We do not consider the initial 20 h period since it is a time over which cells adapt to the starvation conditions in the device, nor the data after 80 h since most cells are dead.

### Initial damage and cell-cycle phase do not correlate with lifespan in most cells

We asked whether initial conditions, namely the cell state when loaded onto the chip, might explain the variations in lifespan (Fig. 2a). There was a negative correlation between initial damage and lifespan (Spearman $r = -0.41$, $p < 0.001$). This correlation was primarily due to a subset of 3% of the cells that had high initial damage (PI uptake rate >4,

compared to the mean uptake rate of 0.87 in the remaining cells). These initially damaged cells had a short lifespan, averaging 48 h.

We therefore divided the cells into two populations, with initial uptake rate above and below 4 (Fig. 2b, c), which we call the high damage and low damage groups. The high damage group showed a strong correlation between initial damage and lifespan (Spearman $r = -0.70$, $n = 17$, $p = 0.002$). The low damage group, which comprised 97% of the cells, showed low correlation (Spearman $r = -0.15$, $n = 503$, $p = 0.001$).

We also investigated the effect of cell-cycle phases by noting the initial size of the cell and number and timing of reductive divisions on the chip[18]. We find that cell size has only weak correlations with lifespan (Spearman $r = -0.09$) (Fig. 2d), as did the time of last division (Spearman $r = -0.11$) (Fig. 2e) and number of divisions (Spearman $r = -0.06$).

Multiple regression shows that initial conditions explain a total of 27% of the variation for all cells, and 9% of the variation for the majority - 97% of the cells - with low initial damage (Fig. 2f). We conclude that in the traits measurable in this experiment, the initial conditions explain only a minority of the variation in lifespan.

### Damage dynamics rise and fall suggesting a stochastic mechanism

We next sought to characterize the stochastic dynamics of damage, defined as PI uptake rate, over time. Damage in each cell did not accumulate monotonically. Instead, damage rose and fell in each cell, with fluctuations larger than can be explained by experimental noise (Fig. 1f). This indicates that damage is produced and removed on the timescale of hours. These fluctuations occurred around a mean trajectory that accelerated with age on the scale of tens of hours. This suggests two timescales: in addition to the fast timescales of hours, a slower timescale of tens of hours over which damage production and removal rates change.

Notably, we find that cells become more similar in relative terms as they age. Although the mean damage and its standard deviation both rise with age (Fig. 3a, b, f), the standard deviation rises more slowly than the mean. As a result, the relative variation drops with age, as measured by the coefficient of variation CV = SD/mean (Fig. 3c). 1/CV rose approximately linearly with age above 50 h.

The increasing relative similarity between cells with age is seen also in the damage distributions at each timepoint. At early ages the distribution is skewed to the right, but skewness reduces with age (Fig. 3d), as the distribution becomes more symmetric. The longitudinal nature of the data allowed us to calculate the autocorrelation of damage. Correlation time increased with age. This means that a cell with damage above or below the population average remained so for longer at old ages (Fig. 3e). Plotting damage as a function of remaining lifetime shows that $X = \ln(\text{normalized PI uptake})$ becomes less dispersed the closer the cell is to death (Fig. 3g).

These findings indicate that the damage dynamics has aspects that become more deterministic with age.

### Damage dynamics indicate a saturated-repair stochastic model

To elucidate the stochastic mechanism that can give rise to these damage statistics, we modeled damage production and removal with noise. We exploited the separation of timescales in the data, namely the rapid fluctuations of damage around a slowly rising mean trajectory. Therefore, we explored models in which damage is produced and repaired quickly compared to the lifespan, whereas the rates of damage production and removal change slowly with age $t$. Damage removal and production were also allowed to depend on the amount of damage to include the possibility of feedback and saturation effects.

We use as a damage variable $X = \ln(\text{normalized PI uptake})$ to represent the loss of the free-energy barrier posed by the membrane in units of $k_B T$. We consider a general stochastic

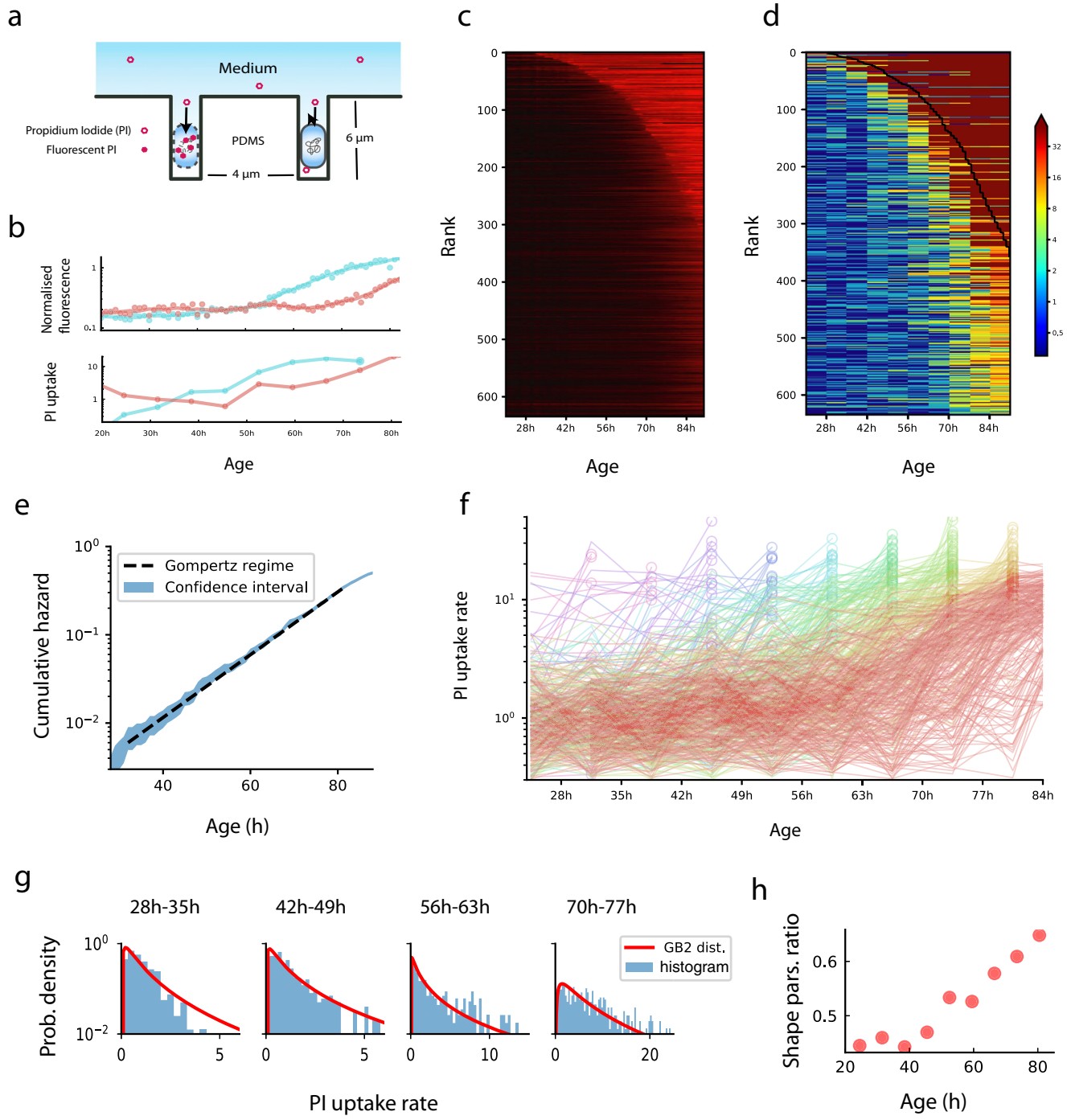

**Fig. 1 | Damage dynamics in starving _E. coli_ cells. a** Individual _E. coli_ cells were placed in microfluidic channels with medium flow. Propidium iodide (PI) added to the medium crosses the membrane and stains DNA when membrane integrity is compromised. **b** Membrane damage was measured by the temporal derivative of PI fluorescence. Shown are, for two individual cells (red and cyan), the fluorescence signals (top) and derived PI uptake rates in 7 h time windows (bottom). **c** Colormaps of normalized fluorescence time-series (background and peak intensities normalized to 0 and 1, colored as black to bright red), with individual cells ranked by lifespan. **d** Colormaps of membrane damage computed from the PI rate of change, with individual cells ranked by lifespan. Solid black line indicates time of death. **e** Cumulative risk of death as a function of age shows an exponential regime.

Cumulative risk of death is defined as negative natural logarithm of survivorship and is equal to the integral of the hazard function. The blue region corresponds to 95% confidence intervals. Death conditions are as previously defined[4]. **f** Cellular damage fluctuates around a rising trajectory, subsampled to 7h time windows. Trajectories are colored according to their time of death (red: after 84 h; yellow: between 77 h and 84 h; yellow-green: between 70 h and 77 h; green, cyan … etc.). Circles indicate the last time window before death. Data shown here are the same as those in (**d**). PI uptake rate is normalized so that the initial timepoints start close to 1 (see Methods). **g** PI uptake rate distributions and best-fit to a type-2 generalized beta distribution with the ratio between shape parameters $p/(p+q)$, plotted versus age in (**h**), see Methods. Source data of (**c**–**h**) are provided in the Source Data file.

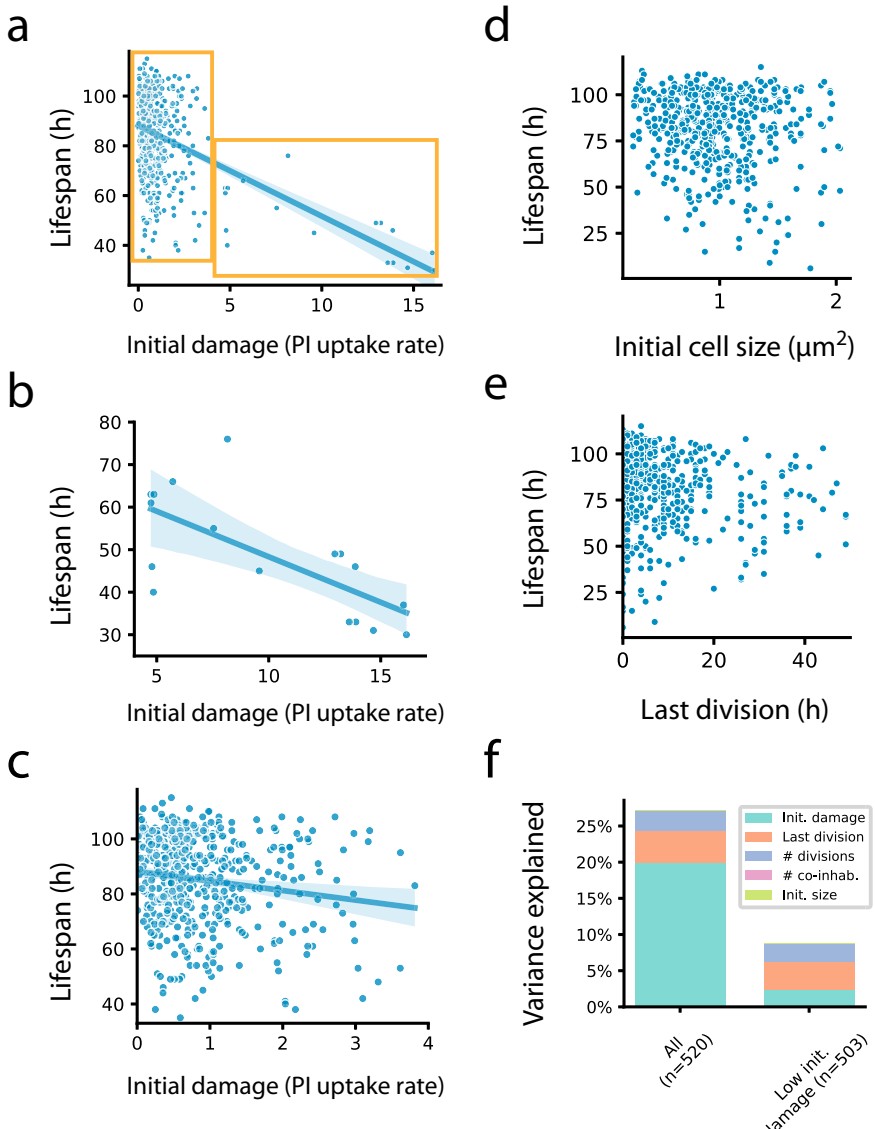

**Fig. 2 | Initial conditions do not account for most of the variations in lifespan.** **a** Initial damage levels (PI uptake rate) and lifespan of all cells in the experiment. Yellow boxes indicate groups of cells with high and low initial damage, each shown separately in (**b**, **c**). **b** For cells with high initial damage (PI uptake rate >4, $n = 17$ cells), initial damage correlates with lifespan. **c** For cells with low initial damage (<4, $n = 503$ cells), the correlation between initial damage and lifespan is weak. **d** Initial cell size and lifespan of all cells in the experiment. **e** Time of last division and lifespan for all cells in the experiment. **f** Fraction of lifespan variation explained by initial conditions according to multiple regression. Left are all cells, right are cells with low (<4) initial damage. Blue lines and shaded regions in panels (**a**–**c**) represent linear regression lines and associated 95% confidence intervals respectively. Source data are provided as a Source Data file.

model $dX/dt$ = production − removal + noise, or mathematically $dX/dt = G(X,t) + \sqrt{2\sigma}\xi$, where $\xi$ is white noise of amplitude σ.

To define the production and removal terms that make up $G(X,t)$, we used timescale separation, by assuming that at each time point the damage distribution among cells $P(X,t)$ is a steady-state solution of the equation. The analytical solution for the steady-state is $P(X,t) = e^{-U(X,t)/\sigma}$, where $U(X,t)$ is a potential function defined by $\partial U/\partial X = -G(X,t)$. This is analogous to the Boltzmann distribution in statistical mechanics.

Using the measured distribution of damage at different time-points, $P(X,t)$, we estimated $U$, differentiated it to provide $G(X,t)$ and hence the production-removal terms in the model.

To facilitate this process, we characterized the experimental damage distributions $P(X,t)$ by comparing them to 15 commonly-used distribution functions with 3–4 parameters (Supplementary Fig. 4, Supplementary Data 1). The best fit for the PI uptake distribution was a type-2 generalized beta distribution[19] with shape parameters whose

ratio, $p/(p+q)$, rises approximately linearly with age (Fig. 1h). The stochastic process which gives rise to this distribution is (see Methods):

$$\frac{dX}{dt} = \eta t - \beta f(X) + \sqrt{2\sigma}\xi \qquad (1)$$

In this inferred mechanism (Fig. 4a) damage production rises linearly with age as $\eta t$, and damage removal is a saturating function of damage, $f(X) = \frac{e^{aX}}{e^{aX} + e^{a\kappa}}$ (Fig. 4b, c). The parameters are a production slope $\eta = (5.1 \pm 0.3) \times 10^{-3} k_B T h^{-2}$ and removal parameters $a = 0.33 \pm 0.02 (k_B T)^{-1}$, $\beta = 1.12 \pm 0.12 k_B T h^{-1}$ and $\kappa = 0.29 \pm 0.07 k_B T$ The white noise amplitude is $\sigma = 0.157 \pm 0.006 (k_B T)^2 h^{-1}$.

Notably, this model is in the same class as the saturated repair (SR) model established for aging in mice[20], in the sense that the production rate of damage rises linearly with age and damage inhibits or saturates its own removal. The only difference is that the mouse SR model used a

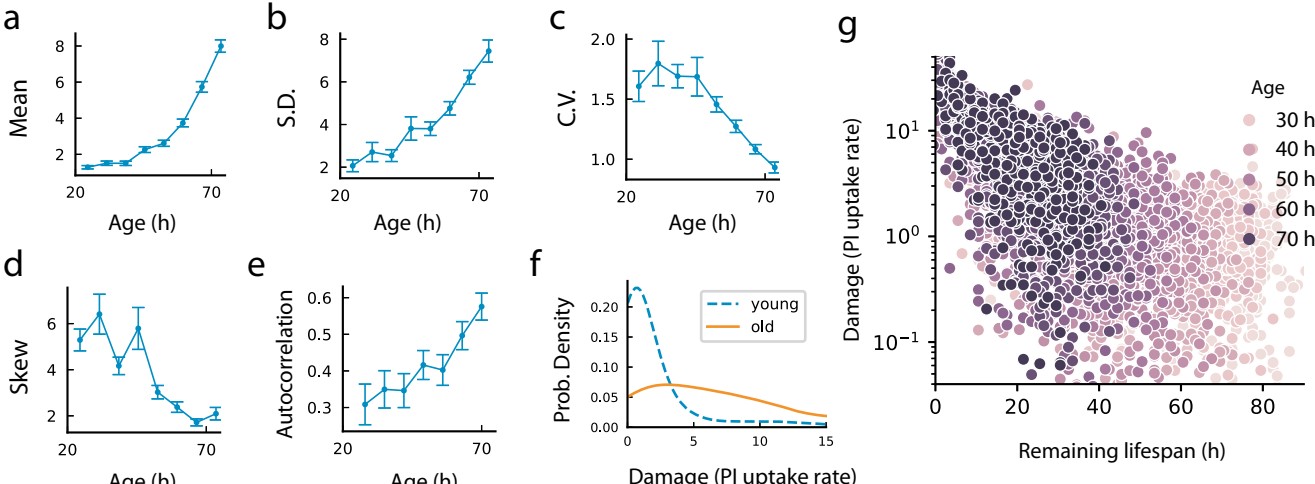

**Fig. 3 | Damage dynamics show increasing determinism with age.** Statistics of *E. coli* membrane damage for all cells alive at a given age (*n* = 635 cells): Mean (**a**) and standard deviation (**b**) increase with age; but coefficient of variation (**c**) decreases, indicating reduced relative heterogeneity in the damage distribution. **d** Skewness drops with age. **e** Autocorrelation of damage (Δt = 7 h) increases with age, showing increasing persistence. All error bars indicate means +/− standard errors estimated from bootstrapping. **f** Probability distribution of damage in younger (52.5 h blue dashed line) versus older (72.5 h yellow solid line) cells. **g** Log PI uptake rate as a function of remaining lifespan becomes less variable close to death. Source data are in the Source Data file.

different saturating removal function, $f(X) = X/(\kappa + X)$. Hence, we call the model of Eq. (1) the membrane-potential-SR model or MP-SR model. We note that the MP-SR removal function $f(X) = \frac{e^{aX}}{e^{aX} + e^{a\kappa}}$ can be interpreted as a two-state partition function that senses the loss of membrane potential $X$.

The MP-SR model captures the statistics of the observed PI uptake dynamics (Fig. 4d–g). It shows a reduction in the relative variation, CV = SD/mean (Fig. 4f), despite a super-linear rise in both mean and SD (Fig. 4d, e). The inverse CV, 1/CV, rises linearly with age as in the data.

The model also captures the reducing skewness with age (Fig. 4g). Hence, the MP-SR model captures the dynamics of damage in the experiment.

To compute the distribution of lifespans in the MP-SR model, we modeled death as damage $X$ exceeding a threshold $X_c$[20]. Death is therefore modeled as a first-hitting-time of the MP-SR model, which we computed numerically and analytically (Supplementary Information) using Kramer's approximation[21,22]. The model provides an exponential increase in the risk of death that slows at very old ages, namely the Gompertz law (Fig. 4h), and a Weibull-like sigmoidal survival curve (Fig. 4i), as experimentally observed. This Gompertzian exponential increase is due in the model to the linear rise in damage production, which causes the potential $U$ to drop linearly with time; since crossing this barrier goes exponentially in $U$, the risk of death rises exponentially with time.

The differences in lifespan between individuals in the inferred stochastic mechanism is due to the fact that noise is effectively amplified by the saturation of damage removal. The slope of production minus removal becomes flat at old ages; fluctuations are not pulled back strongly towards equilibrium by the effective potential $U$ (Fig. 4c). This is at the heart of how noise can generate different lifespans for cells with identical physiological parameters.

We conclude that PI-uptake trajectories and their reducing relative variation are well-explained by an SR-type model in which damage production (loss of membrane barrier function) rate rises linearly with age whereas damage removal saturates.

The SR and MP-SR models make a further prediction that may be called 'shortening twilight'[23,24]. Twilight is the remaining lifespan after a given damage threshold is crossed, and the model predicts that twilight shortens with age. This shortening twilight prediction is borne out by the *E. coli* damage data (Supplementary Fig. 5).

## Dynamics in a strain deleted for stress-response regulation agrees with model predictions

We repeated the experiment with an *E. coli* strain deleted for a master regulator of the stress response, RpoS[25]. Since this strain has reduced stress response, the model makes specific predictions. Reduced stress response should increase the rate of damage production $\eta$ and/or decrease the rate of damage removal $\beta$. These changes in parameters are predicted to result in a shorter lifespan, higher Gompertz slope and, to the extent that $\beta$ is decreased, a shallower survival curve (changes in $\eta$ do not affect survival curve steepness[20]). The model further predicts that damage mean and SD should be higher than the wildtype strain whereas damage CV and skewness should be lower than the wildtype strain.

We tested these predictions using damage measurements from $n = 141$ $\Delta rpoS$ cells in the microfluidic assay (Fig. 5a, b). The data agrees with the model predictions. Lifetime was reduced by 54% (CI: 51–58%) (Fig. 5a), and the Gompertz slope was higher by 56% (CI: 29–89%) (Fig. 5c). The survival curve was only mildly shallower (Fig. 5d), indicating that the parameter β was not strongly affected by the RpoS deletion. Damage mean and SD were higher (Fig. 5e, f), and CV and skewness were lower than the wildtype strain (Fig. 5g, h) as predicted. The findings indicate that the main effect of the RpoS deletion is an increase in the damage accumulation rate parameter $\eta$. The model dynamics with increased $\eta$ and capture the observed dynamics as shown in Fig. 5i, l. We conclude that the model can explain damage dynamics in a strain with reduced stress response.

## Discussion

We studied the role of chance and initial conditions on lifespan by measuring membrane damage over time in starved *E. coli* cells in a microfluidic device. Initial conditions in each cell, such as initial damage, cell size or cell-cycle phase, did not strongly correlate with time of death in most cells. Instead, damage fluctuated in each cell around a rising mean trajectory. Unexpectedly, the relative variation in damage dropped with age. This indicates an increasing determinism with age, where damage levels become more similar in relative terms the older the cells are. Correlation times increased and distributions became less skewed, further indicating rising determinism. The model correctly predicted dynamics in an *E. coli* strain deleted for the stress response regulator RpoS.

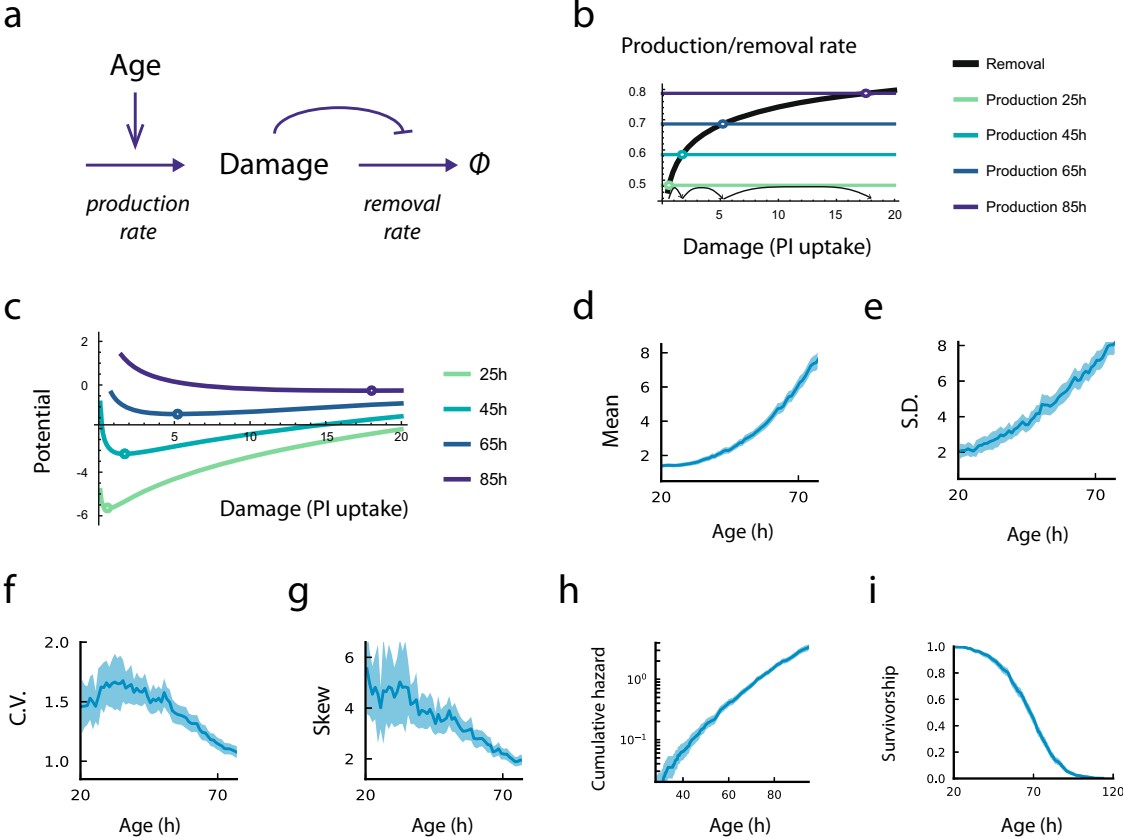

**Fig. 4 | The saturating-removal model captures damage dynamics. a** Schematic of the MP-SR model. **b** Rate plot showing linearly increasing production with age and a removal rate that saturates with damage, causing the fixed point to accelerate to high damage levels. **c** The potential function of the MP-SR model and its evolution with age. Simulations of the MP-SR model for PI uptake rate ($e^X$) show rising mean (**d**) and standard deviation (**e**), reducing CV (**f**) and reducing skewness (**g**).

The model provides a death hazard that rises exponentially with age (**h**) and a Weibull-like survival function (**i**). Blue lines and regions represent the means and 95% confidence intervals respectively from subsampling $n = 6200$ simulated cells. Source data (both simulated trajectories and statistics) are provided in the Source Data file.

We used our dynamical damage measurements to infer a stochastic mechanism that provides the dynamics and survival curves. In this mechanism, damage is produced at a rate that rises linearly with age, and damage-removal saturates at high damage levels. We called the mechanism the membrane-potential saturating repair (MP-SR) model. The MP-SR model predicts well the effects on damage dynamics of deleting the stress regulator RpoS. Our findings suggest that chance fluctuations, amplified by saturating removal of damage, play a major role in explaining why genetically identical bacterial cells in the same conditions die at different times.

The damage dynamics measured here have statistical features that differ from random walks and from most previously suggested models of aging[26–28]. The mean rises faster than the standard deviation, so that the relative heterogeneity between cells at a given age declines. This can be quantitated as a drop in the coefficient of variation, CV = SD/mean, such that 1/CV rises roughly linearly with age. This is an unusual feature in stochastic processes in general, and in previous theoretical models of aging including network models[26,27], the Strehler-Mildvan model[28], the cascading failure model[7], fixed frailty model[8] and Ornstein-Uhlenbeck type models which do not provide a drop in damage CV with age.

The present MP-SR mechanism has two main features that require biological explanations. The first feature is the linear rise with age of the damage production rate, $\eta t$. This linear rise can be explained by assuming that damage arises from 'damage-producing units', such as unfolded-protein complexes, that are added at a constant rate and cannot be resolved or removed[29–35]. If these complexes assimilate new

unfolded proteins at a constant rate, and cannot be removed, their total mass should rise linearly with time. Such unfolded protein complexes are known to be toxic to cells[32]; they cause damage such as dysregulated proteostasis[36], which can lead to membrane damage[37,38].

Mathematically, if cells accumulate damage-producing units $P$ at a constant rate $v$, and these units cannot be removed, their number rises linearly with age, $P = vt$. Each unit produces damage at rate $u$, so that total damage production rate rises linearly with time as $\eta t$ with $\eta = vu$.

Organisms that manage to dilute such damage-producing units $P$, such as organisms with indefinite growth, are predicted to have different damage dynamics, in which $P$ does not rise indefinitely. Such dilution occurs in growing and dividing bacterial cells[32,33], but not in the non-growing starved cells studied here. Other examples of damage dilution may occur in eukaryotic cells with symmetric division such as fission yeast; in contrast, budding yeast with asymmetric divisions show aging and eventual death of the mother cell which retains damage rather than passing it to daughter cells.

The second feature of the MP-SR model is the saturation of damage removal, which is crucial for the present dynamical hallmarks. The relevant removal mechanisms in *E. coli* include chaperones and proteases[36], as well as enzymatic systems that repair proton leakage[39], oxidative damage[40] and maintain membrane structural integrity[41]. Such enzymatic repair mechanisms should naturally saturate at high damage levels.

The inferred stochastic mechanism in *E. coli* is similar to a mechanism inferred in the context of mice aging by Karin et al. Karin et al. used stochastic trajectories of senescent cells in mice, cells which

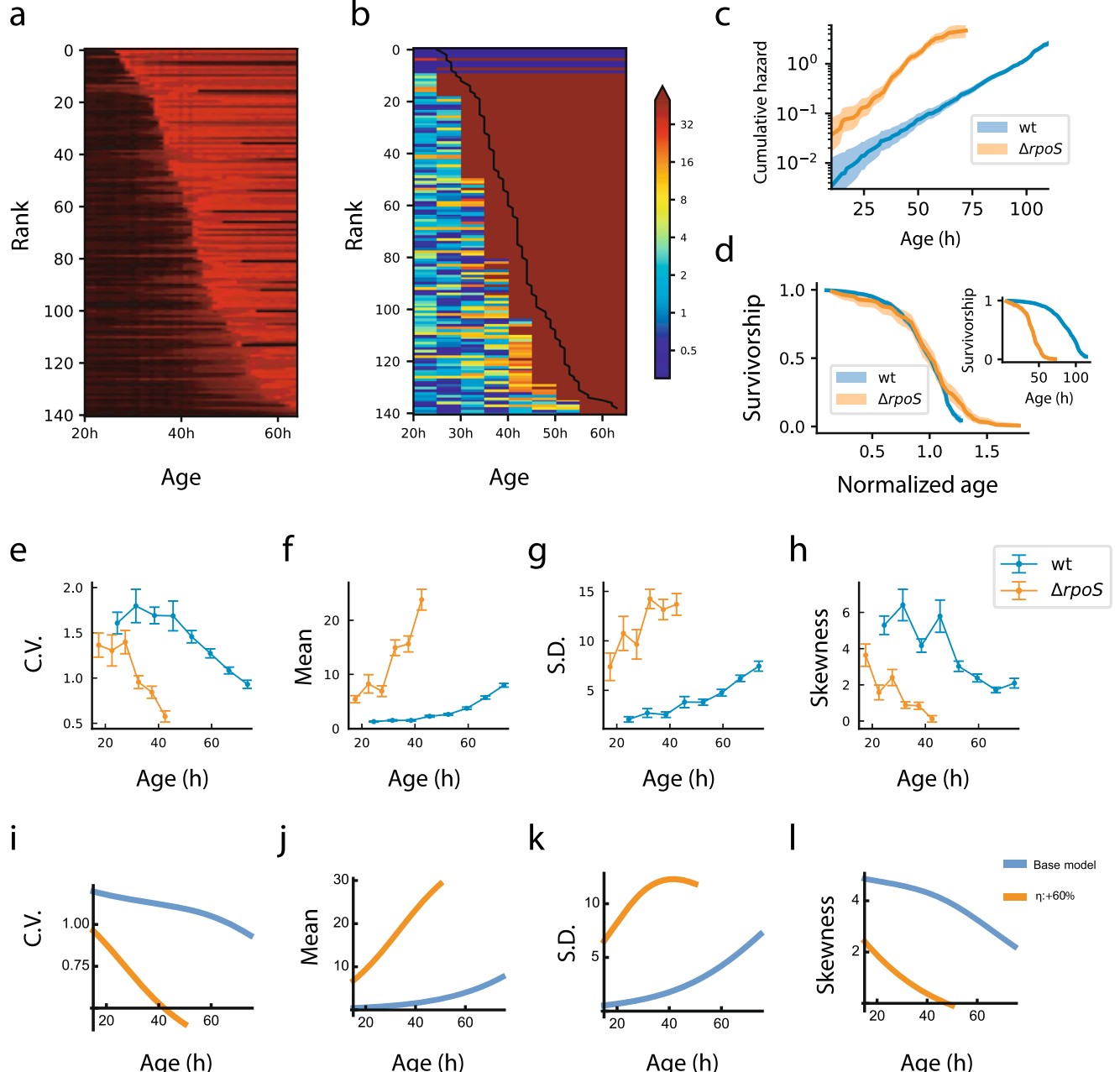

**Fig. 5 | Damage dynamics in a strain that has reduced stress response (ΔrpoS) agree with model predictions. a** Colormaps of normalized fluorescence time-series of ΔrpoS cells (in the same style as Fig. 1c), with individuals ranked by lifespan. **b** Colormaps of estimated PI uptake rates of ΔrpoS cells ranked by lifespan. **c** Cumulative hazard shows increased Gompertz slope. **d** Survivorship shows reduced lifespan (inset), and survivorship versus normalized age shows a mildly shallower survival curve. Solid lines and color bands in panels (**c, d**) indicate means

and 95% confidence intervals respectively. Measured damage statistics (n = 141 cells) include (**e**) CV (**f**) mean, (**g**) SD and (**h**) skewness. Analytical results of MP-SR model with increased $\eta$ show similar dynamics for the (**i**) CV, (**j**) mean, (**k**) SD and (**l**) Skewness. Error bars in panels (**e–h**) indicate means +/− standard errors estimated from bootstrapping. Source data of (**a–h**) are provided in the Source Data file.

are growth arrested cells that cause inflammation, to infer a mechanism for senescent-cell accumulation[20]. This mechanism, called the saturating removal (SR) model, is a stochastic differential equation with a production rate that rises linearly with age and a removal rate that saturates, so that high senescent cell levels slow their own removal. The removal terms in the SR and MP-SR models both saturate; the difference between them may stem from the type of damage: $X$ is senescent cell abundance in the SR model, an extensive variable, whereas $X$ is membrane integrity in the MP-SR model in units of potential, which is intensive and can enter the dynamics in terms of Boltzmann-like factors.

Karin et al. experimentally confirmed a prediction of the SR model, that senescent cell turnover slows with age[20]. The SR model was generalized to other forms of damage, and explains observations on aging such as the Gompertz law, heterochronic parabiosis[42], age-related disease incidence in humans[43] and the scaling of survival curves in *C. elegans*[5,20]. Interestingly, the human frailty index shows similar dynamical features, including a reduction in CV with age[44]. Both the SR and MP-SR model provide shortening twilight[24], in which remaining lifespan after a threshold damage is crossed is reduced with the age.

The similarity between the present study on *E. coli* cells and the model of Karin et al. of mammalian aging hints at a possible

universality in mechanisms of aging, in which chance plays a large role in the differing lifespans of genetically identical organisms. Although the molecular forms of damage and lifespan timescales are different between *E. coli* and mice, the features of linearly rising production and saturating removal may be more general and give rise to similar damage dynamics, with reducing relative heterogeneity with age.

It would be interesting to measure longitudinal damage trajectories in other organisms to explore whether linear-production-and-saturating-removal models might apply more generally. In the context of bacteria, it would be important to explore the dynamics of damage in cells challenged with antibiotics, in order to better understand the role of chance in the function of these drugs.

## Methods

### Microfluidic chip fabrication

The negative master mould for the modified mother machines was fabricated on top of silicon wafers in two steps. First, arrays of dead-end channels ($2000 \times 6\,\mu m$ long) were fabricated via electron-beam lithography (EBL) in a specialized micro-fabrication facility. It was necessary to use EBL for these channels due to the high precision requirements for the cross-section dimensions (both height and width have to be between 1.1 and $1.2\,\mu m$). They had to be large enough to allow single cells to enter yet narrow enough so that multiple cells could not be squeezed in the same channel. In the second step, using standard photolithography methods, the negative mould for the main channel was overlaid perpendicular to the dead-end channels. The main channel is 10 mm long, $50\,\mu m$ wide and $10\,\mu m$ deep.

For each run of the bacterial starvation experiment, microfluidic chips were fabricated by casting PDMS structures out of negative master moulds. Uncured PDMS mixes (RTV-615, Momentive Performance Materials) were poured to a thickness of 3 mm onto the silicon wafer carrying the master moulds, and then de-gassed under vacuuming and spread out via gravity for 2 h. The PDMS was then partially heat cured at 80 °C for an hour to form solid yet flexible PDMS blocks with patterned surfaces. After drilling inlets and outlets through the flow channel, the PDMS blocks were bound to cover glasses suitable for microscopy, using oxygen plasma (90 s, 1000 mTorr). Lastly, the assemblies were cured fully at 80 °C overnight and so that the PDMS structure was sealed permanently to the glass cover slide.

On the day of the experiment, the microfluidic chip was again treated by oxygen plasma for 90 s so that its surfaces were activated, and then injected with 20% (v/v) polyethylene glycol 400 solution for at least 1 h to prevent bacterial adhesion.

### Material and equipment

During the process of media preparation, sterilization, cell culture and fluidic infusion, we generally avoided disposable lab plasticware in favor of glass or equipment whose wetted surfaces are coated with fluoropolymer such as polytetrafluoroethylene (PTFE). This step avoided a pitfall in which trace concentrations of carbon and energy-rich chemicals leached into the media, such as phthalate plasticisers commonly used in PVC tubings. Because in the mother machine we subjected a relatively small number of cells (<10,000) to constantly refreshing volumes (5 µl per hour) of media, such compounds can serve as carbon sources and allow the cells to grow, circumventing the goal of our experiments[4]. Medium was filter-sterilized (0.2 µm) to avoid contamination by volatiles during autoclaving, and glassware was sterilized by dry heat.

### Bacterial growth and loading

All culture media were filter-sterilized before use to remove dust particles, which might otherwise block the microfluidic channels.

*E. coli* wildtype strain MG1655 with a chromosomal-inserted constitutively CFP-expressing cassette (PrrnB2) was grown overnight in M9 minimal media (supplemented by 2 mM $MgSO_4$, and 0.1 mM $CaCl_2$) at 37 °C with 40 mM succinate as carbon source, and diluted 250-fold into 50 ml of the same media in 250 ml Erlenmeyer flasks. This subculture was grown to exponential phase (OD600 ~0.1) at 37 °C and then transferred to glass centrifugation tubes and harvested by centrifugation at 6000 g (Relative Centrifugal Force) for 15 min. The bacterial pellet was resuspended, washed with fresh carbon-free M9 media and centrifuged 3 more times. The resulting pellet was resuspended a final time with 20 µl M9 media. This final suspension was manually injected into the main channel of the microfluidic chip, and forced into the dead-end channels by centrifugation at 1000 rpm for 15 min. After centrifugation, the main channel is washed thoroughly by carbon-free M9 media to remove all cells that remain there.

### Microfluidic and microscopy setup

After the microfluidic chip was loaded with cells, it was connected to a linear, flow-controlled fluidic system driven by a high-precision syringe pump (Harvard Apparatus PHD 2000 Programmable) and GC-grade glass/PTFE syringes (Hamilton Gaslight 1000 series). As mentioned above, PTFE tubing and glass syringes are used to avoid leaching of plasticizers into the media. This is critical for this type of microfluidic starvation experiments, as *E. coli* are able to uptake as carbon sources the trace concentrations of plasticizers in the media when it is constantly refreshed by the microfluidic flow. The syringes are preloaded with filter-sterilized M9 minimal media without carbon source, supplemented with propidium iodide (PI, 5 µg/ml). The chip was first washed at 100 µl per hour for 30 min and then the flow rate was halved every 15 min to a final flow rate of 5 µl per hour. In the meantime, the chip was mounted and stabilized onto the objective stage of an inverted microscope (Nikon ECLIPSE Ti2, 100× oil-immersion objective, controlled with MetaMorph software) with temperature controlled at 37 °C. Phase-contrast and fluorescence (PI signal excitation, 546/12 nm; emission, 605/75 nm; CFP excitation 436/20 nm; emission 480/40 nm) images were automatically taken for up to 90 imaging positions every hour for up to 120 h. Focus was maintained by the hardware-based Perfect Focus System (PFS) from Nikon.

### Image analysis

We used an image analysis method[12] specifically designed for mother machines, implemented as an ImageJ plugin (ImageJ 1.48 v, Java 1.6.0_65 32-bit). The regions in the time-lapse images of each dead-end channel were detected and cut out of the image stacks and displayed chronologically from left to right on the same image (Supplementary Fig. 1b). Cells were segmented using the CFP (constitutively expressed) fluorescent image. The segmentation approach was semi-automatic and consisted of automatic segmentation, lineage assignment and manual correction. First, the central region of the cells was detected using statistical p-value thresholding, assuming that the observed intensities are spatially distributed as Gaussian functions. Then these central regions were used as seeds to add recursively neighboring points with similar intensities to form labeled regions. The result of this automatic process is an accurately segmented image with occasional over segmentation errors. Then labeled regions from different time points in the same dead-end channels were assigned together with arrows to track the same cell through time. These automatically segmented and tracked cells are then manually corrected to account for over-segmentation errors and mis-assignment due to sudden movements of the cells. For each segmented and tracked cell through time, we used the segmented CFP contours to extract the average PI fluorescence signal.

## Measurements of membrane damage

Our general approach is to use the time derivative of fluorescence to calculate the rate constant of PI uptake, which in turn is a proxy for membrane damage.

We model the PI fluorescence time series with one slow and one fast chemical reaction. The slow reaction is PI uptake $PI_{ext} \rightarrow PI_{in}$ with rate constant $r$, and the fast reaction is PI binding to DNA once inside the cell $[PI]_{in} + [DNA] \Longleftrightarrow [PI:DNA]$, assumed reversible and at equilibrium with equilibrium constant $K$, so that $K[DNA][PI]_{in} = [PI:DNA]$.

First, we focus on the rate of PI uptake. The Arrhenius equation states that the logarithm of the rate constant scales linearly with activation energy, in this case, an energetic barrier representing the integrity of the cell membrane. Thus, we defined membrane damage $X(t)$ as the reduction of this energy barrier compared to a healthy baseline. $X(t)$ has the unit of $k_B T$, where $T$ is the experimental temperature 310 K and $k_B$ is the Boltzmann constant. We can choose the unit appropriately, i.e. to be $k_B T$, so that $X(t)$ can be made unitless. Under these definitions, the rate constant is $r = A_0 e^{X(t)}$, where $T$ is the experimental temperature 310 K and $k_B$ is the Boltzmann constant. Then the PI uptake rate is $A_0 e^{X(t)}([PI]_{ext} - [PI]_{in})$.

Having defined the relation between membrane damage and PI uptake rate, our task is the estimation of the latter using fluorescence time-series. Since PI only becomes fluorescent when bound to DNA, average fluorescence intensity is proportional to the bound form of PI: $[Fluo] = J_F[PI:DNA]$. Since the binding of PI to DNA is assumed to be at equilibrium, we have $[PI:DNA] = ([PI]_{in} + [PI:DNA]) \frac{K[DNA]}{1+K[DNA]}$. Thus, the time derivative of fluorescence should be proportional to the PI uptake rate:

$$\frac{d[Fluo]}{dt} = A_0 e^{X(t)}([PI]_{ext} - [PI]_{in}) \frac{J_F K[DNA]}{1+K[DNA]}. \quad (2)$$

To obtain relative fluorescence time-series we normalize for each cell its fluorescence signal $[Fluo]$ by its observed maximum $[Fluo]_{max} = J_F K[DNA][PI]_{ext}$. The relative time series is thus $s(t) = [Fluo]/[Fluo]_{max} = [PI]_{in}/[PI]_{ext}$. Membrane damage can be calculated from the experimentally observed relative fluorescence $s(t)$:

$$\frac{ds(t)}{dt} = \frac{d[Fluo]}{dt}/[Fluo]_{max} = \frac{A_0 e^{X(t)}}{1+K[DNA]}[1 - s(t)] \quad (3)$$

and thus we obtain the formula used in our analysis

$$A_1 e^{X(t)} = \frac{ds(t)/dt}{1 - s(t)}, \quad (4)$$

where $A_1 = A_0/(1 + K[DNA])$.

$A_1$ is an Arrhenius-type pre-exponential factor with a unit of inverse time. It is assumed to be constant, because DNA concentration should be constant among the non-growing cells in our experiment. The value of $A_1$ is not relevant to the dynamics of damage, thus we used $A_1 = 1/600$ so that initial timepoints start close to PI uptake rates of 1.

## Time-series analysis and numerical differentiation

The fluorescence series was zeroed by the background and then divided by the maximum fluorescence for each cell to arrive at $s(t)$ defined above. To arrive at estimates for PI uptake rate, $\frac{ds(t)/dt}{1-s(t)}$, we performed numerical differentiation of $s(t)$ in a fashion that reduces the impact of experimental noise. In the present time-lapse microscopy experiments where single cells inside microfluidic chambers are imaged, experimental noise is driven by fluctuations in focus on the z-axis. This type of noise is approximately multiplicative and non-correlated in neighboring 1 h time points (Supplementary Fig. 2). We therefore smoothed the log-transformed data $\ln[s(t)]$ in time windows of 7 h with a Wiener filter. Then $d\ln[s(t)]/dt$ were estimated using linear regression in non-overlapping 7 h windows. The resulting time derivatives are multiplied by the $s(t)/[1 \cdot s(t)]$ to arrive at $A_1 e^{X(t)}$.

The typical values of $X(t)$ during the lifetime of the bacteria begin around $0.01X_c$ and rise to cross $X_c = 50$ at 80–100 h.

**Marginal damage distributions.** We searched for an analytical form for the probability distribution function that can fit the damage distributions at various ages with age-dependent parameters. We tested 15 commonly used probability distributions. Each distribution has a probability density function $f(Z/b; \Theta)$, where $Z$ is the value of the random variable, $\Theta$ is the vector for the shape parameters and $b$ is the scaling parameter. We fit this to the empirical damage distribution $Z_{it}$ of cell $i$ at age $t$, by maximizing the likelihood $\Sigma_i f(Z_{it}/b_t; \Theta_t)$ as a function of parameters $b_t, \Theta_t$, using the scipy.stats package (version 1.7.3) of python. The goodness of fit was evaluated by the one-sample Kolmogorov–Smirnov (K–S) test. The tested distributions, the K–S test statistics and associated $p$-value are shown in Supplementary Data 1 and Supplementary Fig. 4.

The three distribution functions that best fit the marginal damage distributions, Burr, Burr12 and Fisk, are all special cases of the generalized beta distribution of the second kind (GB2)[19], whose probability density function is:

$$f_{GB2}(Z) = a b^{aq} Z^{ap-1}(b^a + Z^a)^{-p-q}/Beta(p, q), \quad (5)$$

where $p$ and $q$ are dimensionless shape parameters, and $a$ and $b$ describe the cooperativity and scale of the observed damage proxy $Z$, the PI uptake rate. GB2 becomes a Burr (Burr Type III) distribution when $q = 1$ and a Burr12 distribution when $p = 1$, and the Fisk distribution when $p = q = 1$. Since the damage we seek is $X = \ln(Z)$, we transform to obtain:

$$P(X) = f_{GB2}(Z) dZ/dX \sim e^{apX}(b^a + Z^a)^{-p-q}. \quad (6)$$

## Derivation of the MP-SR model

We model the dynamics with a stochastic differential equation (SDE) in the form of

$$dX/dt = G(X, t) + \sqrt{2\sigma}\xi = production - removal + \sqrt{2\sigma}\xi, \quad (7)$$

where both production and removal rates of damage can depend on damage level $X$ and age $t$. We assume that the production and removal of damage happen much faster than the age-related change in parameters. Thus, we can make the approximation that the observed damage distributions in the previous section are quasi-steady-state distributions of the SDE. The quasi-steady-state distribution can be written as the Boltzmann distribution $P(X) \sim e^{-U(X,t)/\sigma}$, where the potential function $U$ is defined by $G(X, t) = -\partial U/\partial X$.

Using the best-fit GB2 distribution $P(X)$ of Eq. (6), we find the potential up to an irrelevant constant:

$$U(X, t) = \sigma(p+q)\ln(b^a + e^{aX}) - \sigma apX. \quad (8)$$

The two terms of this potential function naturally relate to damage production and removal terms. Thus, via differentiation of $U$ with respect to $X$ we find: $production = \sigma ap$, $removal = \sigma a(p+q)e^{aX}/(e^{aX} + e^{aK})$. By redefining the GB2 parameters $b = e^K$, $p = \frac{\eta_t}{a\sigma}$, $q = \frac{\beta_t - \eta_t}{a\sigma}$, the MP-SR model for damage dynamics in *E. coli* is given by:

$$dX/dt = \eta_t - \beta_t \frac{e^{aX}}{e^{aX} + e^{aK}} + \sqrt{2\sigma}\xi. \quad (9)$$

The GB2 parameters that best fit the experimental data show that $p/(p+q)$ rises approximately linearly with time (Fig. 1h) and that $b$ and $a$ remain approximately constant. We conclude that the observed damage distributions are well-described by an SR-type process with $\eta_t = \eta t$ and $\beta_t = \beta$ as in Eq. (1).

**Fitting of the MP-SR model to the damage trajectories**

In order to account for the uncertainty of biological age at the start of the experiments, the model used in fitting the data includes additional constant damage production $\eta_0$. The model equation is $\frac{dX}{dt} = \eta_0 + \eta t - \beta \frac{e^{aX}}{e^{aX} + e^{\kappa X}} + \sqrt{2\sigma}\xi$. We seek the MP-SR model parameters $\hat{\boldsymbol{\theta}} = (\hat{\eta}, \hat{\beta}, \hat{\sigma}, \hat{a}, \hat{\kappa}, \hat{\eta}_0)$ that maximize the likelihood of the observed damage trajectories. For a given parameter vector of the MP-SR model $\boldsymbol{\theta}$, the log-likelihood of the observing the experimental data under this model can be written as $LL(\boldsymbol{\theta}) = \Sigma_i \Sigma_j \ln\{P[(t_j, X_{ij})|(t_{j-1}, X_{i(j-1)}); \boldsymbol{\theta}]\}$, where $P[(t_j, X_{ij})|(t_{j-1}, X_{i(j-1)}); \boldsymbol{\theta}]$ denotes the transition probability of cell $i$ from observed damage of $X_{i(j-1)}$ at age $t_{j-1}$ to $X_{ij}$ at age $t_j$, given by the MP-SR model with parameter $\boldsymbol{\theta}$. Our likelihood maximization algorithm includes simulation-based likelihood calculations and an iterative interval-halving parameter search strategy. At each iteration, in the 6-dimensional parameter space, we set up a grid of parameters at which log-likelihoods are calculated. For each parameter $\boldsymbol{\theta}$ and each of the transitions given by the data ($n = 6651$), we estimate the transition probabilities $P[(t_j, X_{ij})|(t_{j-1}, X_{i(j-1)}); \boldsymbol{\theta}]$ by simulating the MP-SR model with parameter $\boldsymbol{\theta}$ initiating from $(t_{j-1}, X_{i(j-1)})$ to $t_j$ 1000 times using the Mathematica (version 12) function "ItoProcess", and estimate the probability density at $X_{ij}$ by applying the function "Smooth-KernelDistribution". Log-likelihoods are estimated by summing the log transition probabilities, and confidence intervals of log-likelihoods are constructed by bootstrapping the cells. Comparing the log-likelihoods across the parameter grid locates the region of parameter space for the next search iteration. Convergence is reached when (1) optimal parameters are located in the interior of the search grid; (2) log-likelihood differences between the optimal parameter and neighboring alternatives are within confidence intervals. The estimated parameter $\hat{\boldsymbol{\theta}}$ is the parameter that achieves the maximum log-likelihood throughout all iterations. These simulation-based log-likelihood evaluations take a large amount of computing power and are done on a computer cluster. The maximum-likelihood parameters we located for wildtype *E. coli* are $\hat{\eta} = (5.1 \pm 0.3) \times 10^{-3} k_B T h^{-2}$, $\hat{\beta} = 1.12 \pm 0.12 k_B T h^{-1}$, $\hat{\sigma} = 0.157 \pm 0.006 (k_B T)^2 h^{-1}$, $\hat{a} = 0.33 \pm 0.02 (k_B T)^{-1}$, $\hat{\kappa} = 0.29 \pm 0.07 k_B T$ and $\hat{\eta}_0 = 0.36 \pm 0.05 k_B T h^{-1}$.

**Reporting summary**

Further information on research design is available in the Nature Portfolio Reporting Summary linked to this article.

## Data availability

Source data are provided with this paper, including the longitudinal time-series which are the main results of our experiments and the primary data that all subsequent analysis and modeling depends on. In addition, all data and statistics underlying Figs. 1d–h, 2a–e, 3a–e, g, 4d–i, 5e–h, Supplementary Fig. 4 in are included in the Source Data file. All other data are available from the corresponding authors upon request. Source data are provided with this paper.

## Code availability

Custom codes written in Python 3.7 and Mathematica 12 for analysis, statistics and modeling can be found in the following link: https://github.com/y1fanyang/coliDamageDynamics. Image analysis data can also be found in this link.

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

## Acknowledgements

We thank Valery Krizhanovsky for discussions, Nicholas Stroustrup for discussion and sharing data on shortening twilight, David S. Glass for general discussions and editing help. Warm thanks to Chantal Lotton for technical lab support. European Research Council (ERC) grant agreement No 856487 (UA); Axa Foundation Longevity Chair (ABL, PC, INSERM U1284); Bettencourt Schueller Foundation (FBS) (ABL, PC, INSERM U1284).

## Author contributions

Conceptualization: Y.Y., O.K., A.B.L., U.A.; Methodology: Y.Y., O.K., A.M., A.L.S., A.B.L., U.A.; Experiments and data gathering: Y.Y., A.L.S.; Software: X.S; Resources: P.C., A.B.L.; Data curation, Computational investigation: Y.Y.; Formal analysis: Y.Y., A.M., U.A.; Visualization, writing – original draft and writing – review & editing: Y.Y., U.A.

## Competing interests

The authors declare no competing interests.
