## [Peer Review File · Nature Communications]

Damage dynamics and the role of chance in the timing of E. coli cell deathREVIEWER COMMENTS

Reviewer #1 (Remarks to the Author):

In this manuscript, Yang et al. investigated why isogenic cells under the same stress conditions die at different times. This is a longstanding, fundamental question in our understanding of stress resistance and aging. To address this question, the authors quantified the dynamics of damage accumulation in single E.coli cells by measuring the uptake of PI as an indicator of cell membrane damage. It remains challenging to monitor the dynamics of damage accumulation as the direct upstream step to cell death, and this work represents one of the first studies to accomplish that. Based on these measurements, the authors found that the initial conditions or cell cycle phase of cells cannot account for the variation in their final lifespans. Instead, the authors developed a simple mathematical model based on the data, which suggests a linearly increased production of damage coupled with a saturated removal can amplify white noise in the system and account for the observed heterogeneity in lifespan. Specifically, the model is comprised of an age-dependent linear increase in the damage production rate and a damage removal rate that can be saturated by the damage level. This model can nicely capture the experimental data and impressively reproduce a series of nontrivial, characteristic statistics of aging dynamics. In addition, both the data and model reveal a surprising drop in the cell-to-cell variation of damage level with age, indicating that damage levels become relatively more uniform among aged cells preceding death.

I find this work convincing and thought-provoking. The effort toward a unified model of aging is fundamental and important. Therefore, I recommend the acceptance of this manuscript for publication with minor revisions.

Below are a couple of minor concerns:

1. The damage removal term in the MP-SR model is notably different from that in the previous SR model for mice aging by Karin et al. Although both can be saturated, they differ in the kinetics - one is sigmoidal (MP-SR) whereas the other is Michaelis-Menten (SR). It is straightforward to understand why Michaelis-Menten kinetics can be applied to damage removal, but the sigmoidal kinetics of damage removal is less intuitive. It would be very helpful if additional discussions can be provided. For example, (1) What are the potential biological processes underlying the sigmoidal kinetics of damage removal? (2) Why do bacterial aging and mammalian cell senescence need different damage removal term, whereas the damage production is the same? (3) Do these two damage removal terms have different impacts on "twilight"?
2. It would be helpful to show, in a supplemental figure, the raw data for time-series of PI fluorescence and the deduced PI uptake time traces for all the single cells sorted by lifespan, in heatmaps. This will help readers to better appreciate the experimental results that the model is based on.

Nan Hao, PhD
Professor of Molecular Biology, UCSD

Reviewer #2 (Remarks to the Author):

This is a fantastic study by the group of Uri Alon that addresses a very fundamental and unsolved problem in biology: Is cellular aging and death predetermined or subject to stochasticity? The authors have beautiful experimental evidence to support their claims that as cells age, stochasticity becomes less dominant, and cell-to-cell variation in lifespan diminishes with age. It would be great if the authors could perhaps provide just a few additional experiments. I also like to point out that these requests are not "requirements", but rather suggestions to provide a little more data to help readers

and strengthen their work.

Specifically,

1. Unless I missed something, the authors use only Propidium iodide to quantify membrane integrity. It would be great to have one set of measurements with another fluorescent dye, such as Sytox (another cell death reporter). I am not asking the authors to repeat all the experiments using Sytox, just to provide some control that the PI results are consistent with another well-established dye.

2. The authors present a model that explains their observations. It would be great if the authors could perform a "forward experiment" to test if their model can also predict the outcome of a perturbation (new data that was not used to inform the model). For example, what would happen if the cells are exposed to a toxin (an antibiotic or radicals etc)? Since the authors use a microfluidic device, it seems straightforward to expose cells to an agent and determine the effect and compare the outcome to modeling predictions.

3. Along these lines, how does temperature or presence of certain nutrients in the growth media effect their results? Increasing temperature could speed up the aging and also reduce the stochasticity. I thought it would be great if the authors could experimentally reduce the stochasticity in cells and determine its effect on aging and the "shortening twilight".

4. Could the authors please include some images of cells? Since they use a microfluidic device that is optically accessible, it would be helpful for readers like me see just a few actual images of cells in the device and at different time points of the experiment.

Dear Editors,

Thank you for the reviewer comments and for your positive consideration of our manuscript. We were delighted that the reviewers found the work important and convincing, and that they valued the new insights it carries to the understanding of cell aging and death.

We have now addressed the reviewers comments. In particular we add new data that repeats the experiment on a mutant strain with reduced repair, new control data with a different viability stain and additional microscopy images, movies and data heat maps.

We detail our point-by-point responses below in blue.

We believe these revisions make the manuscript stronger, clearer and more rigorous.

We detail our point-by-point responses below.

Sincerely,

Uri Alon and Yifan Yang

Response to reviewer 1:

In this manuscript, Yang et al. investigated why isogenic cells under the same stress conditions die at different times. This is a longstanding, fundamental question in our understanding of stress resistance and aging. To address this question, the authors quantified the dynamics of damage accumulation in single E.coli cells by measuring the uptake of PI as an indicator of cell membrane damage. It remains challenging to monitor the dynamics of damage accumulation as the direct upstream step to cell death, and this work represents one of the first studies to accomplish that. Based on these measurements, the authors found that the initial conditions or cell cycle phase of cells cannot account for the variation in their final lifespans. Instead, the authors developed a simple mathematical model based on the data, which suggests a linearly increased production of damage coupled with a saturated removal can amplify white noise in the system and account for the observed heterogeneity in lifespan. Specifically, the model is comprised of an age-dependent linear increase in the damage production rate and a damage removal rate that can be saturated by the damage level. This model can nicely capture the experimental data and impressively reproduce a series of nontrivial, characteristic statistics of aging dynamics. In addition, both the data and model reveal a surprising drop in the cell-to-cell variation of damage level with age, indicating that damage levels become relatively more uniform among aged cells preceding death.

I find this work convincing and thought-provoking. The effort toward a unified model of aging is fundamental and important. Therefore, I recommend the acceptance of this manuscript for publication with minor revisions.

We thank the reviewer for this warm endorsement.

Below are a couple of minor concerns:

1. The damage removal term in the MP-SR model is notably different from that in the previous SR model for mice aging by Karin et al. Although both can be saturated, they differ in the kinetics - one is sigmoidal (MP-SR) whereas the other is Michaelis-Menten (SR). It is straightforward to understand why Michaelis-Menten kinetics can be applied to damage removal, but the sigmoidal kinetics of damage removal is less intuitive. It would be very helpful if additional discussions can be provided. For example, (1) What are the potential biological processes underlying the sigmoidal kinetics of damage removal?

We thank the reviewer for raising this point, which helped us to clarify the paper.

This comment concerns the biological interpretation of the damage removal term in MP-SR model:

The mathematical form of this term came from the data, specifically the

shape of damage distributions, as described in Method sections “Marginal damage distributions” and “Derivation of the MP-SR model”.

Despite the empirical origin, the functional form of $f(X)$ has a natural physical interpretation: it is the partition function of a two-state system. If the damage repair system has two states, inactive and active, and the free energy differential between the two states depends on the level of damage X , then $f(X) = e^{aX} / (e^{aX} + e^{a\kappa})$ gives the fraction of the repair system in its active state. The exponential terms are interpreted as Boltzmann factors $\exp(\Delta G/kT)$. An example is ATP synthase whose energy states depend on proton gradients.

We accordingly revised the 2nd paragraph of page 9 (in the results section entitled “Damage dynamics indicate a saturated-repair stochastic model”) as follows (changes highlighted) (line 223-229):

Notably, this model is in the same class as the saturated repair (SR) model established for aging in mice²⁰, in the sense that the production rate of damage rises linearly with age and damage inhibits or saturates its own removal. The only difference is that the mouse SR model used a different saturating removal function, $f(X) = X / (\kappa + X)$. Hence we call the model of Eq. 1 the membrane-potential-SR model or MP-SR model. We note that the MP-SR removal function $f(X) = e^{aX} / (e^{aX} + e^{a\kappa})$ can be interpreted as a two-state partition function that senses the loss of membrane potential X .

(2) Why do bacterial aging and mammalian cell senescence need different damage removal terms, whereas the damage production is the same?

The SR model for senescent cells in mice described in Karin *et al* (2019), had a removal term that goes as $f(X)=X/(K+X)$, whereas the present model has $f(X) = e^{aX} / (e^{aX} + e^{a\kappa})$. Both removal terms are saturated by damage. The main reason for the differences in functional forms of the removal term, in our view, has to do with the type of damage X . For mice, X is the amount of senescent cells, which is an extensive variable. In *E. coli*, X measures the loss of membrane integrity in terms of loss potential, which is an intensive variable, and hence Boltzmann-like terms as discussed above appear. It is our view that both models capture essentially the same dynamics.

In the discussion, page 14 paragraph 3, we add (line 359-368):

*The inferred stochastic mechanism in *E. coli* is similar to a mechanism inferred in the context of mice aging by Karin *et al*. Karin *et al* used stochastic trajectories of senescent cells in mice, cells which are growth arrested cells that cause inflammation, to infer a mechanism for senescent-cell accumulation²⁰. This mechanism, called the saturating removal (SR) model, is a stochastic differential equation with a production rate that rises linearly with age and a removal rate that saturates, so that high senescent cell levels slow their own removal. The removal terms in the SR and MP-Sr models both saturate; the difference between them may stem from the type of damage: X is senescent cell abundance in the SR model, an extensive variable, whereas X is*

membrane integrity in the MP-SR model in units of potential, which is intensive and can enter the dynamics in terms of Boltzmann-like factors.

(3) Do these two damage removal terms have different impacts on “twilight”?

The two models do not differ in terms of the “shortening twilight” phenomenon. We are able to reproduce the “shortening twilight” phenomenon in simulations for both models (MP-SR in Figure S5, SR model simulations not included as it is not the subject of the present manuscript). Conceptually, individuals who cross the lower damage threshold at younger ages have smaller damage production term ηt , thus have a longer remaining lifespan. This causes a negative correlation between time to cross the first threshold and remaining lifespan. The mechanistic details of damage removal are not necessary nor relevant to reach this conclusion.

We now modified the SI section entitled “Shortening twilight in the E. coli dataset” (S5) as follows:

We follow the pioneering work of Stroustrup et al and explore the question of twilight, the time between a measurable age-related phenotype to the time of death ⁴⁷. Suppose there is an age-related phenotype that is equivalent to damage crossing a threshold $X1$. If we define twilight ²³ as the remaining lifespan after the threshold is crossed, the question is whether twilight shortens or lengthens with the age at which the threshold is crossed.

The SR and MP-SR models predict that twilight shortens with age on average (Fig. S5ABC). Equivalently, the time to first cross $X1$, denoted t_1 , is positively correlated with time of death t_{\square} (Fig S5D), but with a correlation coefficient less than one (Fig. S5E). This prediction is borne out by the E. coli dataset (Fig. S5FG). A similar effect was observed in C. elegans ⁴⁷.

The reason for shortening twilight in the model is that the damage production term ηt rises with age. Individuals that cross $X1$ at early times have a low production term. It takes them longer (on average) to reach the death threshold than those crossing $X1$ at late times (Fig 1A). Thus there is a negative correlation between t_1 and remaining lifespan (Fig. S5EG).

Figure S5 *E. coli* shows shortening twilight at old age as redacted by the MP-SR model. (A) Examples of two MP-SR model simulation trajectories. Wildtype parameters were used. Thresholds are $XI=10$ for twilight onset and $Xc=20$ for death. Triangles and squares symbolize first-passage times (t_1, t_d) to cross the XI and Xc respectively. (B) Durations of health (t_1 , shown in red) and twilight ($t_d - t_1$, shown in yellow) for 2000 simulated cells ranked by lifespan. (C) Fraction of lifespan spent in health and twilight for the cells, ranked by fraction of time in health. (D) Time to first cross the two thresholds is correlated with slope less than 1 (Regression line $y=0.39x+62.9$) in MP-SR simulations. (E) Remaining lifespan drops with time to cross threshold 1 in MP-SR simulations. (F) *E. coli* lifespan versus the time to cross a damage threshold of normalized PI uptake rate=6 (Regression line is $y=0.89x+23.3$). (G) Remaining lifetime versus time to cross a damage threshold of normalized PI uptake rate =6.

2. It would be helpful to show, in a supplemental figure, the raw data for time-series of PI fluorescence and the deduced PI uptake time traces for all the single cells sorted by lifespan, in heatmaps. This will help readers to better appreciate the experimental results that the model is based on.

We thank the reviewer for this suggestion and we have added to figure 1 heat maps of the raw PI data and the membrane damage computed from the PI rate of change. The new panels are Fig 1 CD.

Figure 1. Damage dynamics in starving *E. coli* cells. (A) Individual *E. coli* cells were placed in microfluidic channels with medium flow. Propidium iodide (PI) added to the medium crosses the membrane and stains DNA when membrane integrity is compromised. (B) Membrane damage was measured by the temporal derivative of PI fluorescence, as shown for two individual bacteria. Top: fluorescence signal, bottom: derivative (uptake rate) in 7h time windows. (C) Colormaps of normalized fluorescence time-series, with individual cells ranked by lifespan. (D) Colormaps of membrane damage computed from the PI rate of change, with individual cells ranked by lifespan. (E) Cumulative risk of death as a function of age shows an exponential

regime. Cumulative risk of death is defined as negative natural logarithm of survivorship and is equal to the integral of the hazard function. The blue region corresponds to 95% confidence intervals. Death conditions are as previously defined⁴. **(F)** Cellular damage fluctuates around a rising trajectory, subsampled to 7h time windows. Trajectories are color coded by the time window of cell death, circles indicate the last time window before death. Data shown here are the same as those in **(D)**. **(G)** PI uptake rate distributions and best-fit to a type-2 generalized beta distribution with the ratio between shape parameters $p/(p+q)$, plotted versus age in **(H)**, see *Methods*.

Response to reviewer 2:

This is a fantastic study by the group of Uri Alon that addresses a very fundamental and unsolved problem in biology: Is cellular aging and death predetermined or subject to stochasticity? The authors have beautiful experimental evidence to support their claims that as cells age, stochasticity becomes less dominant, and cell-to-cell variation in lifespan diminishes with age. It would be great if the authors could perhaps provide just a few additional experiments. I also like to point out that these requests are not “requirements”, but rather suggestions to provide a little more data to help readers and strengthen their work.

We thank the reviewer for this warm endorsement.

Specifically,

1. Unless I missed something, the authors use only Propidium iodide to quantify membrane integrity. It would be great to have one set of measurements with another fluorescent dye, such as Sytox (another cell death reporter). I am not asking the authors to repeat all the experiments using Sytox, just to provide some control that the PI results are consistent with another well-established dye.

We have now added to the revised manuscript control experiments with a second viability stain, AFH+TOPRO. This combined reagent stains both damaged proteins (AFH) and DNA (TOPRO), and thus extends our main dye PI which only stains DNA. We find good agreement between lifespan measured by PI in our microfluidic experiment and conventional FACS assay at a single time point of fraction surviving using AFH+TOPRO.

We now add this to a new SI section S3, quoted below:

Lifespan based on PI agrees with viability using a different viability stain, AFH+TOPRO.

We compared the present lifespan measurements using PI to viability using a different stain, AFH+TOPRO⁴⁵. This combined reagent stains both damaged proteins (AFH) and DNA (TOPRO), and thus extends our main dye PI which only stains DNA. We find good agreement between lifespan measured by PI in our microfluidic experiment and conventional single time-point (day 7) FACS assay of fraction surviving using AFH+TOPRO (Fig. S3)

*We used PI in our experiments because of previous validation in *E. coli*^{4,46} as a non-toxic dye to track cell viability longitudinally. Longitudinal experiments have different requirements than single-time-point experiments on cell viability - for example, uniformity of maximum intensity across cells might be more important in the latter. Longitudinal analysis allows imaging transients and end-stages for each cell, and thus avoids some of the concerns inherent in single-time point studies.*

In addition, our time-lapse microscopy experiments also capture not only the time course of the PI signal, but also the image of each cell throughout time (Figure S1, Movie S1). Thus we are able to confirm that the increases of PI signals are concentrated inside of the cells and centered around the nucleoids. These observations rule out the possibility of extracellular nucleic acids reported to occur in biofilms.⁴⁶

Figure S3 Lifespan measurements using PI in microfluidic experiments correlate well with single-time point longevity measurements using AFH+TOPRO in batch culture. Each marker in orange corresponds to one *E. coli* strain (all based on MG1655 backgrounds). For each strain, we have performed microfluidic experiments ⁴ similar to those performed in the present study, in order to measure their survival curves. Plotted on the x-axis are median lifespan of these strains in microfluidic experiments. We also performed conventional batch-culture experiments to test cell viability, in conditions similar to those in microfluidics, and tested cell viability with AFH+TOPRO staining and FACS. Each strain was tested three times. Plotted on the y-axis are the average fluorescence signal at day 7, normalized by cell size (estimated by forward scatter).

Fig. S3 include the lifespan measurements of the following strains, all based on MG1655 backgrounds: wildtype, Δ agaA, Δ agaR, Δ agaV, Δ alsB, Δ appA, Δ appB, Δ appY, Δ aqpZ, Δ ascG, Δ bglF, Δ bipA, Δ caiB, Δ caiC, Δ caiT, Δ caseE, Δ cbpM, Δ citT, Δ crcA, Δ cstA, Δ cynX, Δ dcp, Δ dcuA, Δ dnaT, Δ eptA, Δ eutQ, Δ fabR, Δ fixC, Δ fkpA, Δ flgD, Δ flu, Δ fruB, Δ fruR, Δ fsr, Δ glf, Δ glpX, Δ gltD, Δ gltK, Δ gltL, Δ gnsA, Δ gspF, Δ gspL, Δ gspM, Δ hslR, Δ hycA, Δ idnD, Δ intG, Δ ldcC, Δ lsrC, Δ lysC, Δ mela, Δ mhpB, Δ mprA, Δ mtlD, Δ nagK, Δ nagZ, Δ napC, Δ nirC, Δ norV, Δ norW, Δ nsrR, Δ ompL, Δ osmY, Δ paaC, Δ paoB, Δ pfkA, Δ pflC, Δ pflD, Δ pgi, Δ phnH, Δ ptsH, Δ rarD, Δ recF, Δ rph, Δ rsd, Δ sgcQ, Δ sspA, Δ ssuA, Δ ssuE, Δ syd, Δ tdcE, Δ thiQ, Δ treC, Δ tyrB, Δ ulaD, Δ uxaA, Δ uxaB, Δ uxaC, Δ uxuR, Δ xdhB, Δ xdhC, Δ yafV, Δ yafX, Δ yagH, Δ yagL, Δ yagQ, Δ yagS, Δ yajQ, Δ ybaJ, Δ ybbP, Δ ybdG, Δ ybdK, Δ ybiT, Δ ycaD, Δ ycaL, Δ ycfX, Δ ydcP, Δ ydgT, Δ ydjK, Δ ydjQ, Δ yecP, Δ yeeW, Δ yehA, Δ yfbM, Δ yfiD, Δ yfjO, Δ ygcQ, Δ ygfO, Δ yggE, Δ yggF, Δ yggP, Δ yhbP, Δ yhbS, Δ yhbW, Δ yhdX, Δ yiaG, Δ yiaW, Δ yich, Δ yicM.

2. The authors present a model that explains their observations. It would be great if the authors could perform a “forward experiment” to test if their model can also predict the outcome of a perturbation (new data that was not used to inform the model). For example, what would happen if the cells are exposed to a toxin (an antibiotic or radicals etc)? Since the authors use a microfluidic device, it seems straightforward to expose cells to an agent and determine the effect and compare the outcome to modeling predictions.

We thank the reviewers for this comment that helped us to add new data on a perturbation. Specifically, we hypothesized that the stress response system plays a role in damage production and removal and thus studied an E coli strain deleted for a major stress-response regulator *rpoS*.

The model suggests that $\Delta rpoS$ should have reduced damage repair (both for damage producing units and for the rapid damage X), and thus have increased η (more damage producing units such as unfolded protein complexes due to lack of chaperones and proteases) and/or reduced β (less removal of X). With these changes to MP-SR model parameters, we predict the following changes to the observed damage statistics:

- 1) Shorter median half life ($\sim \beta/\eta$)
- 2) Higher mean and SD of damage;
- 3) Reduced CV and skewness of damage.
- 4) shallower survival curve (steepness $\sim 1/\beta$) or, if primarily η is affected, survival curve of similar steepness (scaled survival curve);

We provide the new data and statistics of the deleted *RpoS* strain in a new Fig. 5 for $n=138$ individual cells in the microfluidic assay. The *RpoS* data matches the model predictions, with median lifetime that shortens by about 51-58%. The data suggests that η grows by around 50-60%:

We write in results in a new section entitled “Dynamics in a strain deleted for stress-response regulation agrees with model predictions”:

*We repeated the experiment with an E coli strain deleted for a master regulator of the stress response, *RpoS*²⁵. Since this strain has reduced stress-response, the model makes specific predictions. Reduced stress response should increase the rate of damage production η and/or decrease the rate of damage removal β . These changes in parameters are predicted to result in a shorter lifespan, higher Gompertz slope and, to the extent that β is decreased, a shallower survival curve (changes in η do not affect survival curve steepness²⁰). The model further predicts that damage mean and SD should be higher than the wildtype strain whereas damage CV and skewness should be lower than the wildtype strain.*

*We tested these predictions using damage measurements from $n=138$ $\Delta rpoS$ cells in the microfluidic assay (Fig. 5AB). The data agrees with the model predictions. Lifetime was reduced by 54% (CI: 51%-58%) (Fig. 5A), and the Gompertz slope was higher by 67% (CI: 38%-102%) (Fig. 5C). The survival curve was only mildly shallower (Fig. 5D), indicating that the parameter β was not strongly affected by the *RpoS* deletion. Damage mean and SD were higher (Fig 5EF), and CV and skewness were lower than the wildtype strain (Fig. 5GH) as predicted. The findings indicate that the main effect of the *RpoS* deletion is an increase in the damage accumulation rate parameter η . The model dynamics with increased η and capture the observed dynamics as*

shown in Fig 5 I-L. We conclude that the model can explain damage dynamics in a strain with reduced stress response.

Figure 5. Damage dynamics in a strain that has reduced stress response ($\Delta rpoS$) agree with model predictions. (A) Colormaps of normalized fluorescence time-series of $\Delta rpoS$ cells, with individuals ranked by lifespan. (B) Colormaps of estimated PI uptake rates of $\Delta rpoS$ cells ranked by lifespan. (C) Cumulative hazard shows increased Gompertz slope, (D) Survivorship shows reduced lifespan (inset), and survivorship versus normalized age shows a mildly shallower survival curve. Measured damage statistics include E) CV F) mean, G) SD and H) skewness. Model simulations with increased η show similar dynamics for the I) CV, J) mean, K) SD and L) Skewness.

3. Along these lines, how does temperature or presence of certain nutrients in the growth media effect their results? Increasing temperature could speed up the aging and also reduce the stochasticity. I thought it would be great if the authors could experimentally reduce the stochasticity in cells and determine its effect on aging and the “shortening twilight”.

We agree that the effects of temperature and nutrients are fascinating. However, such perturbations require lengthy new calibration and would delay publication significantly. Such perturbations tend to be pleiotropic and would be difficult to predict their effects a priori. We feel therefore that these measurements are outside the scope of the current study.

4. Could the authors please include some images of cells? Since they use a microfluidic device that is optically accessible, it would be helpful for readers like me see just a few actual images of cells in the device and at different time points of the experiment.

We have now added images of cells together with segmentation time courses in a new SI section (below). We also add a time-lapse movie to the SI. We believe that this helps to clarify the experimental procedure.

Figure S1 Sample microscopy images (A) and demonstration of the image analysis process explained in the relevant method section (B).

[See Movie file submitted alone with the manuscript]

Movie S1 Time-lapse movie of one sample imaging position. Each frame is a pseudo-color image, color-merged from CFP and PI fluorescence images.

In summary, the reviewer comments helped us to add new data and controls. We believe the revised manuscript is much improved in terms of strength, rigor and clarity.

REVIEWERS' COMMENTS

Reviewer #1 (Remarks to the Author):

The authors have nicely addressed all my concerns. I therefore recommend the acceptance of this manuscript for publication.

Nan Hao, PhD
Professor of Molecular Biology, UCSD

Reviewer #2 (Remarks to the Author):

The authors have done a great job addressing my comments.
I have no further comments and strongly endorse publication of this exciting study.